# Health and Sustainability in Public Meals—An Explorative Review

**DOI:** 10.3390/ijerph17020621

**Published:** 2020-01-18

**Authors:** Karin Höijer, Caroline Lindö, Arwa Mustafa, Maria Nyberg, Viktoria Olsson, Elisabet Rothenberg, Hanna Sepp, Karin Wendin

**Affiliations:** 1Faculty of Natural Sciences, Kristianstad University, SE- 291 88 Kristianstad, Sweden; 2Dept. of Food Science, University of Copenhagen, DK-1958 Copenhagen, Denmark

**Keywords:** health, sustainability, public meals, Europe

## Abstract

The world is facing a number of challenges related to food consumption. These are, on the one hand, health effects and, on the other hand, the environmental impact of food production. Radical changes are needed to achieve a sustainable and healthy food production and consumption. Public and institutional meals play a vital role in promoting health and sustainability, since they are responsible for a significant part of food consumption, as well as their “normative influence” on peoples’ food habits. The aim of this paper is to provide an explorative review of the scientific literature, focusing on European research including both concepts of health and sustainability in studies of public meals. Of >3000 papers, 20 were found to satisfy these criteria and were thus included in the review. The results showed that schools and hospitals are the most dominant arenas where both health and sustainability have been addressed. Three different approaches in combining health and sustainability have been found, these are: *“Health as embracing sustainability”*, *“Sustainability as embracing health”* and *“Health and sustainability as separate concepts”*. However, a clear motivation for addressing both health and sustainability is most often missing.

## 1. Introduction

The world is facing a number of challenges related to food consumption. On the one hand, health effects related to lifestyle factors such as dietary habits, may lead to non-communicable diseases (NCD) causing the deaths of 41 million people each year, equivalent to 71% of all deaths globally, and also malnutrition with 462 million people being underweight [1,2]. Of the six WHO regions, Europe is the most severely affected by Non-Communicable Disease (NCDs), the four major NCDs—cardiovascular disease, diabetes, cancer, and respiratory diseases—together account for 77% of the burden of disease and almost 86% of premature mortality. Excess bodyweight and intake of energy, saturated fats, trans fats, sugar, and salt, as well as low consumption of vegetables, fruits, and whole grains are considered the leading risk factors [3]. According to malnutrition the demographic situation in Europe with a growing proportion of the population researching ever-increasing ages means increasing prevalence of disease and thereby increasing risk of disease-related malnutrition [4,5]. On the other hand, the environmental impact of food production and distribution is currently responsible for about 25–30% of total greenhouse gas emissions (GHGE) [6] and radical changes are called for to achieve a target of a maximum 2 °C t increase [7,8]. For the EU food and drink are responsible for 20–30% of various environmental impacts derived from private consumption when analysing the life cycle for all goods consumed within EU [9].

Taken together, there is an urgent need to integrate health and environmental aspects both at an individual and a societal level, in order to achieve sustainability in food consumption in accordance with the United Nation’s (UN) sustainability goals (SDGs) [10]. The public/institutional meal plays an important role in striving for this goal [11], not only because it is responsible for a significant part of many people’s food consumption, but also through its normative influence on peoples’ food habits [12]. In Sweden alone, about 3 million public meals are served per day at an estimated cost of 20–25 billon SEK per year. These meals are regulated by a number of laws, such as the Educational Act, Social Services Act and the Public Procurement Act according quality and financing.

In this paper, *public meals* are defined as meals taking place in institutional settings [11,13,14]. While the sustainability perspective has only recently been introduced as a research subject in the public meal arena, the health perspective is well established [8,15].

The primary aim of this study is, therefore, to provide an explorative review of the scientific literature, focusing on European research including concepts of health and sustainability in studies of public meals.

The specific research questions are:-In which public meal arenas has the combination of health and sustainability been addressed?-Which aspects of health and sustainability are primarily put forward?-How are health and sustainability associated and what are the motives for combining these factors?

## 2. Background

### 2.1. Public Meals

Public meals appear in different arenas and in different forms, covering various needs [16]. The areas of focus in Sweden, as well in western society in general, have often been rational efficiency, and health and nutrition [17]. There are no common regulations for how public meals should be designed or financed. However, using the school lunch as an example, many countries have policies to provide nutritionally balanced meals reflecting the general food culture of the country [18,19,20].

Sweden is one of the leading countries in Europe in terms of the number of public meals served daily and per capita [21]. There is a long history of serving meals free of charge in schools and preschools that started in 1945, and from 1974 free school meals have been served in all Swedish municipalities. The school meal was identified in Sweden early on as an arena for health actions as well as a way to achieve equality in health [22], reaching not only the children but also their parents. These meals have been shown to improve the diet of children [23,24]. In Figure 1, sectors serving public meals in Sweden are shown as an example, illustrating the large variety of public meal settings.

Studies have also indicated the role of meals in relation to the health of patients in hospitals [25] and older adults in care homes [26]. These meals constitute one of the most basic parts of medical treatment with the aim of meeting specific nutritional requirements in relation to various medical conditions [27]. In this aspect, patient safety is of the utmost importance with regard to the quality of meals served in the healthcare context. In addition, there are sensory demands; the meals should be appetizing and also constitute an arena for social interaction. Studies have been conducted focusing on food and meals in relation to health among inmates in prison, where the problems of obesity and an unhealthy diet have been raised [28,29]. A study by Wangmo et al. [30] acknowledged the importance of nutritional interventions in prison aimed at improving health. Only a few studies have focused on food and meals in the military service as a public meal arena [31,32].

### 2.2. Aspects of Health and Sustainability

Both “health” and “sustainability” are complex concepts and their meanings are not always consistent. Already in 1946, the World Health Organization (WHO) created a definition of health that aimed not only at focusing on the physical aspects but also the social and mental aspects of health. Even though this definition has been recurrently criticized for being too utopian, it is still referred to and has been awarded for its holistic approach to health.

“*Health is a state of complete physical, mental and social well-being and not merely the absence of disease or infirmity*” [36].

The role of a healthy diet for the growth of children and the maintenance of health and prevention of disease is indisputable [15,37]. In relation to the social and mental aspects of health, studies have indicated that shared meals, and especially family meals, have positive effects [38,39]. Some studies have also investigated the role of the school meal on mental health, including levels of stress among the students [31], as well as the role of meals for well-being in care homes for elderly persons [40].

In 1983, the United Nations created the World Commission on Environment and Development (Brundtland Commission), which defined sustainable development as:
“*Meeting the needs of the present without compromising the ability of future generations to meet their own needs*” [10].

In 1992, the first United Nations Conference on Environment and Development (UNCED) was held in which Agenda 21 was developed and adopted. When preparing for a follow up meeting, Rio+20 2012, Colombia proposed the idea of the SDGs, which was adopted by the United Nations Department of Public Information, 64th NGO Conference in Bonn, Germany. The resulting 17 SDGs and associated targets were adopted by all United Nations Member States in 2015. Many of the goals included health and the environment aspects.

The EAT-report [8] from 2019 states:
“*Without action, the world risks failing to meet the UN Sustainable Development Goals (SDGs) and the Paris Agreement, and today’s children will inherit a planet that has been severely degraded and where much of the population will increasingly suffer from malnutrition and preventable disease*”.

## 3. Materials and Methods

An explorative review was conducted where scientific databases were searched for articles discussing the public meal in relation to both health and sustainability. The search string used is reported below and the used databases are given in Table 1. The criteria for selection of papers are given in Table 2.

Search string for identification of relevant literature:

(“Public meal” OR “public meals” OR “Institutional meal” OR “Institutional meals” OR “School meal” OR “School meals” OR “pre-school meal” OR “pre-school meals” OR “School lunch” OR “School lunches” OR “Workplace meal” OR “Workplace meals” OR “Institutional food” OR “Institutional catering” OR “Public restaurant” OR “Public restaurants” OR “Public sector meal” OR “Public sector meals” OR “Public food service” OR “Public food services” OR “School meal system” OR “School meal systems” OR “Public kitchen” OR “Public kitchens” OR “Food service” OR “Food services” OR “institutional fare” OR “Catering service” OR “catering services” OR catering OR Canteen OR Canteens) AND (Sustainability OR Sustainable) AND (Wellbeing OR Health).

20 papers fulfilled the criteria and were thus included in this review (Table 3). These papers were further analysed for the following information:Aim;Public meal arena;Aspects of health;Aspects of sustainability;Motive for combining health and sustainability.

The selected articles and their characteristics are compiled in Table 3 and Table 4.

## 4. Results and Discussion

The primary aim has been to provide an explorative review of the scientific literature focusing on European research, where both health and sustainability in relation to public meals have been addressed. In total, 20 articles were analysed, and these are presented in Table 3.

By connecting Table 3 to Table 4 using the reference number, each reference may be easily identified. As shown in Table 4, the main public meal arena for discussing health and sustainability was the school, including preschool. School was the main arena in a total of 12 articles, a result also in line with previous studies, where the school has been the primary public meal arena investigated [61].

Our findings show that social and mental aspects of health were found to be addressed in only a few articles, e.g., in terms of social equality and overall well-being and growth. In for example paper No 2 school meal programs increase knowledge and awareness of norms around sustainable consumption to meet challenges in both health and sustainability. In papers No 11 and 19 school meals are said to be a tool to improve health in children across ethnic and socioeconomic groups. (Table 2). Commonly, health was understood in terms of physical health only, focusing on a healthy diet and pointing to the urge to eat according to national dietary guidelines. It was illustrated, for example in papers No 3, 4 and 11, as intake of fruit and vegetables, other nutritious food, and an adequate energy intake, often in the context of overweight and obesity among children.

Furthermore, sustainability was primarily dealt with in relation to environmental aspects, for example in paper No 5, 13, 14, 17 and 18; however, social and economic aspects were also mentioned in some of the selected articles, see No 1, 2, 3, 4, 6, 7, 10, 11, 15, 19 and 20. As in earlier studies [62,63], in selected papers for example No 11 and 13, sustainability was most often discussed in terms of food waste, but procurement and local production were also discussed in relation to both environmental and economic aspects in papers No 5, 10 and 15. The domination of the environmental aspect of sustainability has previously been criticized [64]. However, in this review, social and economic aspects were in papers No 6 and 10, to some extent, included primarily in terms of politics, welfare, and social justice, and to highlight the importance of including relevant actors (Table 4).

The study showed the different approaches to health and sustainability as well as ways of combining these aspects. Three main approaches could be identified, which were evident in some studies as exemplified below, and less apparent in others. The approaches identified are:-**Health as embracing sustainability**, where health is the point of departure and where sustainability is included as part of health. This is emphasized in relation to health promotion initiatives and how these could also be more sustainable, claiming that health should embrace both aspects. This is the content of papers No 14 and 19 where health and nutritional aspects of school meals also would include sustainability in the form of agricultural improvements or effectiveness. In papers No 16 and 17 health promotion in canteens may promote sustainability through environmental thinking and behaviours. Paper No 3 claims that increased consumption of fruit and vegetables as a part of a healthy diet will have a positive impact on sustainability.-***Sustainability as embracing health,****where sustainability is in focus and where health should be seen as part of sustainability*. This was for example illustrated when focusing on sustainable food procurement which is then also motivated by better nutrition in terms of knowing where the food comes from and how it is produced in papers No 5, 10 and 15 This is also exemplified in studies were environmental challenges are in focus, which thereby requires new sources of nutrition and healthy food. In these cases, health is considered as part of the concept of sustainability. Examples of this are paper No 1 and 6, where food consumption in schools is in focus and paper No 7, where restaurant meals were studied. In paper No 13 infant food and food waste were focused upon.-***Health and sustainability as separate concepts***, where the link between them was unspecified or undefined. This could be exemplified by the stated role of the school meal to tackle societal challenges related to health and sustainability, although separately which is seen in paper No 8, 9, 10, 12 and 20 In papers No 2, 4, 18 meals in schools and preschool were used as a pedagogical tool, *the learning role* of the meal was highlighted in these studies, which included learning about both health and sustainability. For example, when investigating portion size both health, in terms of obesity and overweight, and sustainability, related to food waste, were identified as important factors.

Most often, a clear motivation for addressing or associating both concepts was lacking. When motivations were given, the need to include challenges related to both health and sustainability was put forward as them both relating to food behavior. This is clear in paper No 12 in which the role food scape is discussed in terms of impact and understanding of food behavior, but also in paper No 11 where school lunch was studied. Thereby, these studies also achieved a more holistic perspective. Another motive for combining health and sustainability was to gain a deeper understanding of the complexity of food behavior, which is seen in papers No 14, 19 and 20. According to the SDGs, WHO Non-communicable diseases country profiles 2018 [2], the UN Intergovernmental Panel on Climate Change (IPCC) [10] and the Lancet EAT report [8], there is an urgent need to increase scientific knowledge about how to embrace both health and sustainability aspects in the context of public meals.

Public meals play an important role in influencing people’s food behavior as well as maintaining or improving health [25,26]. Although public meals follow us through most stages of life, from cradle to grave, they have been more or less scientifically overlooked [65]. During recent decades, the focus on sustainability has increased in public debate and policymaking, also putting public meals on the agenda as an important means of achieving both health benefits and sustainability in society [66,67]. Earlier research has generally taken the health aspects of catering in public meals as the point of departure, while sustainability has not been a focus. Since the SDGs were formulated in 2015, a more integrated approach has been put forward. The fact that food and meals are related to most SDGs supports the need for further research and practical efforts directed toward sustainability in the area of public meals.

Based on the large number of meals served in public settings every day, these meals have the potential to target both health and sustainability challenges in relation to food and meals. It can be pointed out that the result of this study indicates an *association* rather than an *integration* between health and other concepts in all included articles, but especially in paper No 3, 5, 9, and 20. Additionally, health in terms of nutritional needs did not necessarily imply a positive environmental impact as discussed in paper No 10.

## 5. Concluding Discussion

It may be concluded that schools and hospitals are the most dominant arenas where both health and sustainability have been addressed in order to reach a more holistic perspective on food consumption of public meals. Three different approaches in combining health and sustainability have been found, which are:
“*Health as embracing sustainability”, “Sustainability as embracing health” and “Health and sustainability as separate concepts*”.

However, a clear motivation for addressing both health and sustainability is most often lacking.

This review includes 20 articles, a rather low number explained partly by the European focus, which can be seen as a limitation. However, even if articles from outside Europe had been found, the non-European context may have been confusing in terms of public meals since, at present, a common definition of public meals is lacking. Another explanation is that the research area of sustainability is quite new, with the 17 UN SDGs only being adopted in 2015. Furthermore, the focus on school/pre-school meals could be regarded as a limitation, but from the literature search, it was obvious that this is the most well-documented arena in terms of public meals.

The strength of the present study is its aim to explore studies covering the aspects of health and sustainability in the context of public meals at the same time. To our knowledge, this is the first paper with this focus. In conclusion, indicating an urgent need for research within all public meal arenas, between which conditions and challenges may vary, according to issues of health and sustainability. An increased number of publications also opens opportunities for systematic review analyses allowing the findings of separate papers to be compared and contrasted while providing a foundation for decision-making.

## Figures and Tables

**Figure 1 ijerph-17-00621-f001:**
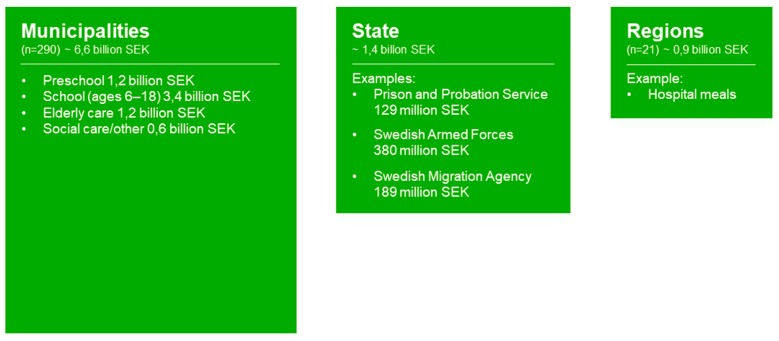
The figure illustrates different sectors serving meals to users. The sizes of the boxes correspond to the number of meals served in each arena, based on a Swedish context [33,34,35].

**Table 1 ijerph-17-00621-t001:** Databases for search of papers.

Academic Search Premier
Cinahl Complete
Education Research Complete
ERIC
Medline
PubMed
SAGE Journals Online
Taylor & Francis Online
Web of Science
Wiley Online Library
Scopus

**Table 2 ijerph-17-00621-t002:** Criteria for inclusion of papers.

Written on Studies and by Researchers Based in Europe
Based on title and abstract, cover both the perspectives health and sustainability
Concern with public meals.
Full text and in English
Peer reviewed
Published 2008–2018

**Table 3 ijerph-17-00621-t003:** The 20 articles on health and sustainability in public meals fulfilling the five criteria for inclusion in this study.

No.	Reference
No 1.	Jones, M.; Dailami, N.; Weitkamp, E.; Salmon, D.; Kimberlee, R.; Morley, A.; Orme, J. Food sustainability education as a route to healthier eating: evaluation of a multi-component school programme in English primary schools. *Health Educ. Res.* 2012, *27*, 448–458 [41].
No 2.	Oostindjer, M.; Aschemann-Witzel, J.; Wang, Q.; Skuland, S.E.; Egelandsdal, B.; Amdam, G.V; Schjøll, A.; Pachucki, M.C.; Rozin, P.; Stein, J.; et al. Are school meals a viable and sustainable tool to improve the healthiness and sustainability of children’s diet and food consumption? A cross-national comparative perspective. *Crit. Rev. Food Sci. Nutr.* 2017, *57*, 3942–3958 [42].
No 3.	Jones, M.; Pitt, H.; Oxford, L.; Bray, I.; Kimberlee, R.; Orme, J. Association between Food for Life, a whole setting healthy and sustainable food programme, and primary school children’s consumption of fruit and vegetables: a cross-sectional study in England. *Int. J. Environ. Res. Public Health* 2017, *14*, 1–15 [43].
No 4.	Balzaretti, C.M.; Ventura, V.; Ratti, S.; Ferrazzi, G.; Spallina, A.; Carruba, M.O.; Castrica, M. Improving the overall sustainability of the school meal chain: the role of portion sizes. *Eat. Weight Disord. Stud. Anorex. Bulim. Obes.* 2018, 1–10 [44].
No 5.	Saxe, H.; Loftager Okkels, S.; Jensen, J. How to obtain forty percent less environmental impact by healthy, protein-optimized snacks for older adults. *Int. J. Environ. Res. Public Health* 2017, *14*, 1–21 [45].
No 6.	Wickramasinghe, K.; Rayner, M.; Goldacre, M.; Townsend, N.; Scarborough, P. Environmental and nutrition impact of achieving new School Food Plan recommendations in the primary school meals sector in England. *BMJ Open* 2017, *7*, 1–7 [46].
No 7.	Engelmann, T.; Speck, M.; Rohn, H.; Bienge, K.; Langen, N.; Howell, E.; Göbel, C.; Friedrich, S.; Teitscheid, P.; Bowry, J.; et al. Sustainability assessment of out-of-home meals: potentials and challenges of applying the indicator sets NAHGAST meal-basic and NAHGAST meal-pro. *Sustainability* 2018, *10*, 1–22 [47].
No 8.	Gray, S.; Orme, J.; Pitt, H.; Jones, M. Food for Life: evaluation of the impact of the Hospital Food Programme in England using a case study approach. *JRSM Open* 2017, *8*, 1–9 [48].
No 9.	He, C.; Mikkelsen, B.E. The association between organic school food policy and school food environment: results from an observational study in Danish schools. *Perspect. Public Health* 2014, *134*, 110–116 [49].
No 10.	Sonnino, R.; McWilliam, S. Food waste, catering practices and public procurement: A case study of hospital food systems in Wales. *Food Policy* 2011, *36*, 823–829 [50].
No 11.	Thorsen, A.V.; Lassen, A.D.; Andersen, E.W.; Christensen, L.M.; Biltoft-Jensen, A.; Andersen, R.; Damsgaard, C.T.; Michaelsen, K.F.; Tetens, I. Plate waste and intake of school lunch based on the new Nordic diet and on packed lunches: a randomised controlled trial in 8-to 11-year-old Danish children. *J. Nutr. Sci.* 2015, *4* [51].
No 12.	Mikkelsen, B.E. Images of foodscapes: Introduction to foodscape studies and their application in the study of healthy eating out-of-home environments. *Perspect. Public Health* 2011, *131*, 209–216 [52].
No 13.	Ryan-Fogarty, Y.; Becker, G.; Moles, R.; O’Regan, B. Backcasting to identify food waste prevention and mitigation opportunities for infant feeding in maternity services. *Waste Management* 2017, *61*, 405–414 [53].
No 14.	Nelson, M.; Breda, J. School food research: building the evidence base for policy. *Public Health Nutr.* 2013, *16*, 958–967 [54].
No 15.	Filippini, R.; De Noni, I.; Corsi, S.; Spigarolo, R.; Bocchi, S. Sustainable school food procurement: What factors do affect the introduction and the increase of organic food? *Food Policy* 2018, *76*, 109–119 [55].
No 16.	Doherty, S.; Cawood, J.; Dooris, M. Applying the whole-system settings approach to food within universities. *Perspect. Public Health* 2011, *131*, 217–224 [56].
No 17.	Thorsen, A.V.; Lassen, A.D.; Tetens, I.; Hels, O.; Mikkelsen, B.E. Long-term sustainability of a worksite canteen intervention of serving more fruit and vegetables. *Public Health Nutr.* 2010, *13*, 1647–1652 [57].
No 18.	Pittman, D.W.; Parker, J.S.; Getz, B.R.; Jackson, C.M.; Le, T.A.; Riggs, S.B.; Shay, J.M. Cost-free and sustainable incentive increases healthy eating decisions during elementary school lunch. *Int. J. Obes.* 2012, *36*, 76–79 [58].
No 19.	Moore, L.; de Silva-Sanigorski, A.; Moore, S.N. A socio-ecological perspective on behavioural interventions to influence food choice in schools: alternative, complementary or synergistic? *Public Health Nutr.* 2013, *16*, 1000–1005 [59].
No 20.	Gray, S.; Jones, M.; Means, R.; Orme, J.; Pitt, H.; Salmon, D. Inter-sectoral Transfer of the Food for Life Settings Framework in England. *Health Promot. Int.* 2017, *33*, 781–790 [60].

**Table 4 ijerph-17-00621-t004:** Main content of the included articles.

Ref No	Aim	Public Meal Arena	Aspects of Health	Aspects of Sustainability	Motivation for Combining Health and Sustainability
No 1. [41]	To examine the associations between promotion of sustainable food issues in primary schools and student self-reported fruit and vegetable consumption and associated student behaviors.	School	Food sustainability may have impact on re-energizing multi-component health programs in schools as a conceptually coherent set of practices.	*Social* justice and cultural regeneration may create alternative routes for health education. *Environmental*/Climate change impact of food cannot be addressed unless individuals and communities know how the food is produced and regain skills and knowledge to take control over what they eat.	Ecological, ethical and welfare aspects of food as a foreground and as part of the global debate about food security and the environmental impacts of an industrialized food system.
No 2. [42]	To highlight research on the history and health implications of school meal programs in a cross-national comparative framework. Specifically to discuss the current role of school meals as a tool for improving food behaviors and population health in a sustainable way.	School	School meal programs contribute to teaching children their culinary heritage and norms around consumption, sustainability, and health in their resident country, and they can create a social and physical learning environment around food that may help to tackle current challenges in health and sustainability.	*Social:* School meals may improve health and impact food choice because of the time devoted to eating in schools, the potential to form new food habits and the importance of the school social environment. *Environmental:* Food production impacts biodiversity, climate change, and imbalances of the nitrogen and phosphorus cycle, in particular via food waste, water usage, and greenhouse gas emissions in farming, and through food processing and transportation.	No clear definitions or disagreements on what healthy and sustainable mean will be challenging during any attempt to evaluate impact of school meals.
No 3. [43]	To examine the association between primary school engagement in the Food for Life programme (FFLP) and the consumption of fruit and vegetables by children aged 8–10 years.	School	Intake of fruits and vegetables as part of a healthy diet	*Social:* Fair-trade certified produce and high animal welfare standards. Engagement with food producers and the local community. *Environmental:* Sustainable food defined as in-season, local and organic produce, marine conservation certified fish and diets high in fruit and vegetables.	No definition or explanation.
No 4. [44]	To analyze food meal portions in primary schools and their compliance with standard portions.	School	Children’s overweight/obesity	Concerns of *social* implications of the challenge of finding innovative solutions to the increasing overweight/obesity rates in children. Food loss and waste have negative *environmental* impacts because of the water, land, energy, and other natural resources used to produce the wasted food. Food waste has a negative impact on the *economy.*	Food portion size is important for both overweight and food waste.
No 5. [45]	To investigate environmental impact of snack recipes with the aim of improving sustainability in public procurement.	Elderly Care	Sufficient energy and protein intake.	*Environmental* impact of snack servings in public procurement can be reduced by 40%.	Both have significant socioeconomic implications.
No 6. [46]	To estimate expected changes in the nutritional quality and greenhouse gas emissions (GHGEs) of primary school meals due to adoption of a new mandatory food-based standard for school meals.	School	Measurements to identify nutritional quality or healthiness of a primary school meal. Two nutrient-based definitions were used to define a ‘healthy school meal’. First: based on saturated fat, non-milk extrinsic sugars and salt. Second: based on 14 nutrients used by the PSFS to quantify nutritional quality of school meals.	Improvement in ‘healthiness’ of a meal does not automatically guarantee a positive *environmental* impact. The opposite can be seen in many cases.	Improving both the nutritional quality and environmental impact of diets is difficult. GHGE reduction is a priority concern globally which requires actions from all sectors. Both production and consumption of food need to be addressed. Tackling both climate change impact and health impacts of food are important policy priorities globally. Looking at the GHGE of diets and their nutritional quality or health impacts is of importance.
No 7. [47]	To develop an assessment tool for measuring the sustainability of a meal/recipe.	Restaurant	Health was included in sustainability and consisted of figures describing the amount of: energy, fiber, carbohydrate, sugar and salt.	*Social* indicators are included in the tool by shares of fair-trade products Negative impact on *environment* caused by nutrition/health should be limited. *Economic* indicators include popularity and cost-coverage ratio.	Health is a part of the sustainability.
No 8. [48]	To evaluate impact and challenges of implementation of a Food for Life approach within three pilot sites.	Hospital	No definition or explanation.	No definition or explanation.	Hospitals need support and clearer performance measures in the area of food. Models of centralized cook-chill meals for patients may offer opportunities to influence specifications to improve nutrition, quality and sustainability.
No 9. [49]	To examine the influence of organic food sourcing policies on the development of healthier school food environments.	School	The Food and Nutrition Policy is a set of principles that aims to fulfil the nutritional needs of pupils and ensure availability and accessibility of healthy foods in schools.	No definition or explanation.	Experience suggests that a rethink of school meals seems to be based on the perspectives: organic sourcing and healthy eating.
No 10. [50]	To address the need for more comprehensive studies on sustainable food systems through a case study of hospital food waste.	Hospital	No definition or explanation	*Social*: The study raises the need for a more integrated political approach. *Environmental*: Food waste management should be a part of a sustainable development of food systems. *Economical*: Public procurement could be part of the complex system and development of sustainable food systems.	Sustainable food procurement is about nutrition, where the food comes from, how it is produced and transported, and where it ends up. It is also about food quality, safety, and choice. Meaning that it is about defining the best value in a broad sense.
No 11. [51]	To investigate whether the amount of food intake and total and relative edible plate waste differed between packed lunches from home and school meals. A further objective was to examine how food intake and food waste are associated with the liking of meals.	School	Health equals eating in accordance with the national dietary guidelines. Schools are important settings to improve access to healthy foods for preventing overweight, obesity and chronic diseases in the long term and to reach children across all ethnic and socioeconomic groups.	*Social*: A more politically integrated approach is needed to mobilize all actors in the food system around a shared vision for sustainable development. *Environmental*: New Nordic Diet dietary principles include: palatability, environmentally friendly, and based on food originating from the Nordic region. Another key principle is to reduce waste.	Awareness of food waste in a dietary sustainability context.
No 12. [52]	To explore the notion of foodscapes and discuss its relevance in understanding determinants of food behavior in institutional out-of-home eating environments. A further aim is to contribute to the development of foodscape studies as a tool to understand and develop health behavior while eating in out-of-home environments.	Restaurant	Food environment impacts our behavior and health, the ‘agency’ of the physical environment and assumes that the environment can act in ways that are both supportive and counterproductive for a given food behavior.	*Social:* A foodscape can comprise values related to the local economy, poor primary producers in the global south and/or animal welfare *Environmental:* A foodscape can comprise values related to the environment and biodiversity.	Food production and consumption has significant implications on our health as well as on the environment. Foodscapes may capture change agendas related to healthier and more sustainable production and consumption.
No 13. [53]	To evaluate the use of back casting as a tool for mitigation of Ready to Use breast-milk substitute food waste.	Hospital	Health is related to nutrition for infants.	Sustainability in terms of energy demand and the negative *environmental* effects of food waste in hospitals.	Based on climate change and the environmental impact of food consumption, new sources of nutrition are needed. The ambition of reducing waste from infant formula feeding might lead to increased breastfeeding which is suggested to support the health of the infant.
No 14. [54]	To outline a rationale for school food research, monitoring, and evaluation in relation to policy, and to identify ways forward for future working.	School	Improvements in healthy food and good nutrition, but also well-being and growth, by schools to promote the health of the children. Nutrition strategies in schools are discussed in terms of the ability to help both children and their families to become empowered in consuming and promoting healthy and sustainable diets.	Impact of school meals on the *environment* both inside and outside the school.	School food and nutrition may provide a cohesive core for health, education and agricultural improvement.
No 15. [55]	To examine the public food procurement system in municipalities of Lombardy in Northern Italy.	School	Discussed in terms of healthy eating.	*Environmental*, *social* and *economic* aspects of sustainability should be considered.	Points at the interrelation between healthy diets, economic profitability, social inclusion, and environmental impact concerning sustainable school food procurement.
No 16. [56]	To discuss how the Healthy Universities approach can help to ensure a holistic and integrated approach to addressing issues related to food.	Canteen	Health promotion.	No definition/explanation.	Addresses a range of issues that make up the university ‘foodscape’ and thereby promote health in an integrated way taking into account the relationships between environment and behavior.
No 17. [57]	To analyze the 5-year sustainability of a worksite canteen intervention of serving more fruit and vegetables.	Canteen	Health promotion.	*Environment* refers to worksites.	Health promotion can be made more sustainable by targeting interventions to specific settings (worksites).
No 18. [58]	To create an intrinsic reinforcement of pride and self-esteem through sustainable, extrinsic public recognition when students were allowed to ring a call bell in the cafeteria in front of their peers after choosing the identified healthiest lunch items with non-flavored milk.	School	Food choice has an impact on health and may be guided by the food pyramid guide.	*Environment* equals the place where you eat, in this case, a cafeteria.	A cost-free and sustainable practice to increase healthy eating decisions during elementary school lunch service without substantial changes to the routine or environment.
No 19. [59]	To highlight the potential importance of viewing alternative approaches as complementary or synergistic, rather than competing, to improve the dietary intake of schoolchildren.	School	Refers to Whitehead taxonomy of multiple levels of influence on health. eg. (i) individual lifestyle factors; (ii) social and community networks; (iii) living and working conditions; and (iv) socioeconomic, cultural and environmental conditions. Further to the socio-ecological health promotion framework where human development is shaped by systems or contexts.	*Social* marketing can be used to target decision-makers and implementation staff with the intent of having them influence the social determinants of health and social inequalities. Social marketing can be used to target audiences by ‘selling’ the personal, social and environmental benefits of change. Food choice and consumption are strongly influenced by a range of factors operating at multiple levels including the *environment*. In the socio-ecological framework, environment is a level of change with the target “Policies, advocacy, environments and structures that impact on health”. So, the environment is both physical and the education system. Socioeconomic are important factors.	In addressing childhood obesity in a sustainable way, dietary patterns need to be improved overall. Rather than being seen as competing alternatives, diverse approaches to improving the diets of schoolchildren should be considered in terms of their potential to be complementary and synergistic.
No 20. [60]	To examine the transferability of practice and learning between settings and tease out the role of whole system frameworks for stimulating change due to a health and sustainability program.	School, hospital, care-home	Health as determined by environmental, organizational and personal factors which interact in complex ways.	*Social* aspects in terms of “a positive meal culture”. *Environmental* aspects in terms of sustainable procurement, reduced waste loss, animal welfare.	The intersection between food, health and sustainability is of greatest importance.

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
