# Peer review of "Health and Sustainability in Public Meals—An Explorative Review"

_ijerph, 2020, doi:10.3390/ijerph17020621_

Round 1
Reviewer 1 Report
This paper is a review of selected research from the European literature on the nexus of human health and sustainability in publicly-funded meals, mostly in schools. It does not examine these topics in depth but rather summarizes the 20 papers in a large table and suggests some general groupings based on how the topics are presented and handled. These are topics with growing interest and of greater importance, so in that sense any contributions could be of value to the ongoing conversations about best practices. On the other hand the authors focus on just 20 papers, which seems quite limited for a review of such topics. There is also not much synthesis of the summary information presented in the table.
Specific comments:
Line 19: Why is “normative influence” in parentheses in the Abstract but not in the body of the manuscript?
Line 22 – At this point the reader doesn’t know anything about the criteria that were used to select 20 of the 3000 papers, so telling the reader that the criteria were satisfied doesn’t mean anything.
Lines 34-37: As currently written, the second sentence of the Introduction is misleading because it implies that diseases related to food consumption account 71% of deaths globally. The WHO fact sheet cited states that: “Noncommunicable diseases (NCDs) kill 41 million people each year, equivalent to 71% of all deaths globally.” It also states: “Noncommunicable diseases (NCDs), also known as chronic diseases, tend to be of long duration and are the result of a combination of genetic, physiological, environmental and behaviours factors.” Clearly NCDs are not all food related.
Line 38: The authors state that food production and distribution is responsible for 25% of GHG emissions (globally) but the source they cite states 25-30%. Further, this isn’t a primary source for this information. That cited paper cites three other papers for the information with publication dates from 1999 to 2007. Surely the authors can find a current primary source for this figure (e.g. IPCC).
Line 40: “maximum 2° temperature increase” should be “maximum 2°C increase”
Line 44: One “in” should be deleted.
Line 67: The first sentence of the Background begins with: “In a European context, public meals appear…” Wouldn’t this statement be equally valid outside of Europe?
Line 85: Appears to be an extra space after “meeting”
Line 88: Extra space before “In”
Line 94: Extra space after “arena”
Line 121: Extra space after “up”
Line 122: Sustainable Development Goals (SDGs) needs to be spelled out at first mention.
Line 134: Materials and Methods are usually plural.
Line 135: The entire text for the Material and Methods section amounts to less than 50 words. It appears that the authors have attempted to use a figure in place of text to minimize this section but I see several problems with this approach. 1) The figure is hard to read due to the small font size and organization of the text (or lack of). The first box in particular is just a stream of search terms. Enlarging the figure to 120-150% helps some but the graphics are blurred. 2) Figures are usually there to support text and improve understanding for the reader but in this case the reader is left to interpret a figure with a caption that is just four words long. Typically a figure with its caption should be able to stand alone from the text and still be understood. I don’t think this meets that standard. 3) One last point is that the figure states that there were 21 articles that met the criteria yet only 20 are shown in the tables. This may seem like a minor difference but with only 20 in the sample population adding one more is a 5% increase.
Line 141: I suggest this table caption be: Table 1. The 20 (or 21?) articles on health and sustainability in public meals fulfilling the five criteria for inclusion in this study.
Lines 179-199: Why was there only one example given for each of the three broad categories of articles? Based on the aim and research questions, I would expect the authors would classify all 20 articles within their framework. Is there a reason only one example was given for each?
Line 209: There appear to be several unnecessary spaces to delete in this line.
Author Response
This paper is a review of selected research from the European literature on the nexus of human health and sustainability in publicly-funded meals, mostly in schools. It does not examine these topics in depth but rather summarizes the 20 papers in a large table and suggests some general groupings based on how the topics are presented and handled. These are topics with growing interest and of greater importance, so in that sense any contributions could be of value to the ongoing conversations about best practices. On the other hand the authors focus on just 20 papers, which seems quite limited for a review of such topics. There is also not much synthesis of the summary information presented in the table.
Thank you for the input. The manuscript has been revised to meet this comment.
Specific comments:
Line 19: Why is “normative influence” in parentheses in the Abstract but not in the body of the manuscript?
This is now added to the manuscript.
Line 22 – At this point the reader doesn’t know anything about the criteria that were used to select 20 of the 3000 papers, so telling the reader that the criteria were satisfied doesn’t mean anything.
This is now specified.
Lines 34-37: As currently written, the second sentence of the Introduction is misleading because it implies that diseases related to food consumption account 71% of deaths globally. The WHO fact sheet cited states that: “Noncommunicable diseases (NCDs) kill 41 million people each year, equivalent to 71% of all deaths globally.” It also states: “Noncommunicable diseases (NCDs), also known as chronic diseases, tend to be of long duration and are the result of a combination of genetic, physiological, environmental and behaviours factors.” Clearly NCDs are not all food related.
Now changed and specified.
Line 38: The authors state that food production and distribution is responsible for 25% of GHG emissions (globally) but the source they cite states 25-30%. Further, this isn’t a primary source for this information. That cited paper cites three other papers for the information with publication dates from 1999 to 2007. Surely the authors can find a current primary source for this figure (e.g. IPCC).
Reference is changed.
Line 40: “maximum 2° temperature increase” should be “maximum 2°C increase”
Changed
Line 44: One “in” should be deleted.
Deleted
Line 67: The first sentence of the Background begins with: “In a European context, public meals appear…” Wouldn’t this statement be equally valid outside of Europe?
“In a European context” deleted.
Line 85: Appears to be an extra space after “meeting”
Deleted
Line 88: Extra space before “In”
Deleted
Line 94: Extra space after “arena”
Deleted
Line 121: Extra space after “up”
Deleted
Line 122: Sustainable Development Goals (SDGs) needs to be spelled out at first mention.
This is mentioned already at line 43.
Line 134: Materials and Methods are usually plural.
Done
Line 135: The entire text for the Material and Methods section amounts to less than 50 words. It appears that the authors have attempted to use a figure in place of text to minimize this section but I see several problems with this approach. 1) The figure is hard to read due to the small font size and organization of the text (or lack of). The first box in particular is just a stream of search terms. Enlarging the figure to 120-150% helps some but the graphics are blurred. 2) Figures are usually there to support text and improve understanding for the reader but in this case the reader is left to interpret a figure with a caption that is just four words long. Typically a figure with its caption should be able to stand alone from the text and still be understood. I don’t think this meets that standard. 3) One last point is that the figure states that there were 21 articles that met the criteria yet only 20 are shown in the tables. This may seem like a minor difference but with only 20 in the sample population adding one more is a 5% increase.
The Figure is deleted and information regarding materials and method is transformed into text and tables. The analysis is based on 20 papers.
Line 141: I suggest this table caption be: Table 1. The 20 (or 21?) articles on health and sustainability in public meals fulfilling the five criteria for inclusion in this study.
Done
Lines 179-199: Why was there only one example given for each of the three broad categories of articles? Based on the aim and research questions, I would expect the authors would classify all 20 articles within their framework. Is there a reason only one example was given for each?
The Discussion is expanded to further highlight the results.
Line 209: There appear to be several unnecessary spaces to delete in this line.
Thank you, this has been corrected.

Reviewer 2 Report
I would like to thank you for the opportunity to review the manuscript. I am disappointed. By reading the manuscript entitled "Health and sustainability in public meals - an explorative review" I expected something more than just a table in which I can find information about what other researchers have done. I agree with the authors that this may be the first such article in this direction. But this direction and level is too low for IJERPH. Even for a literature review, this is not enough. There is not even a discussion where the authors would take a confrontation of views. The article should be rebuilt - do not place tables 1 and 2 where there are links to the articles, but present their content, proven hypotheses and results obtained in the text.
Author Response
I would like to thank you for the opportunity to review the manuscript. I am disappointed. By reading the manuscript entitled "Health and sustainability in public meals - an explorative review" I expected something more than just a table in which I can find information about what other researchers have done. I agree with the authors that this may be the first such article in this direction. But this direction and level is too low for IJERPH. Even for a literature review, this is not enough. There is not even a discussion where the authors would take a confrontation of views. The article should be rebuilt - do not place tables 1 and 2 where there are links to the articles, but present their content, proven hypotheses and results obtained in the text.
We are sorry about the disappointment. We have expanded the discussion to include more reflections on the article’s Content.

Reviewer 3 Report
In general, the manuscript "Health and sustainability in public meals - an explorative review" seems valid, although more efforts could have been made to expand this review.
Specific comments
You could add "Europe" in the Keywords.
The Introduction doesn't contain a sufficient background especially at European level (on which research focuses) and therefore needs substantial additions.
The Materials and Method section also needs to be expanded. Figure 2 is not mentioned in the text and also needs to be discussed. In addition, it's barely readable and therefore its quality must be improved.
Results and Discussion: I seem to read in Figure 2 that the selected papers are 21, but in Table 1 there are 20. Please clarify this aspect. Furthermore, it seems unthinkable that from a total of over 3000 papers only 20 were suitable for analysis.
Table 2 should be discussed further, trying to highlight the results obtained.
I suggest you to add the Conclusions with some further personal reflection and integrate Strengths and limitations of this study into them.
Author Response
In general, the manuscript "Health and sustainability in public meals - an explorative review" seems valid, although more efforts could have been made to expand this review.
Specific comments
You could add "Europe" in the Keywords.
Now added.
The Introduction doesn't contain a sufficient background especially at European level (on which research focuses) and therefore needs substantial additions.
We have expanded the Introduction according to European perspective
The Materials and Method section also needs to be expanded. Figure 2 is not mentioned in the text and also needs to be discussed. In addition, it's barely readable and therefore its quality must be improved.
See comments to review 1
KW: Results and Discussion: I seem to read in Figure 2 that the selected papers are 21, but in Table 1 there are 20. Please clarify this aspect.
Thank you for pointing this out. The correct number is 20. Figure 2 is deleted and the information is added as text and tables.
Furthermore, it seems unthinkable that from a total of over 3000 papers only 20 were suitable for analysis.
When performing the search for suitable articles fulfilling the aim of the review, the approach was systematic. Generally the process in systematical reviews starts with a great number of papers, but according to criteria the revew normally ends up with a quite limited number of articles. In comparison to publications where systematic review has been performed the numbers, 3000 and 20, are normal. This since a large majority of the found articles did not fulfill the specific criteria for this review.
Table 2 should be discussed further, trying to highlight the results obtained.
The Discussion is expanded to further highlight the results.
I suggest you to add the Conclusions with some further personal reflection and integrate Strengths and limitations of this study into them.
Personal reflections are integrated in the text.

Round 2
Reviewer 1 Report
It appears that all issues have been addressed.
Author Response
Reviewer 1
R: It appears that all issues have been addressed.
A: Thank you for this comment!

Reviewer 2 Report
After re-examining the text, I would like to say that it has its advantages. A very good review article would emerge from this manuscript.
I do not understand why the authors insist on presenting the views of other researchers in tabular form.
This presentation would be very nice in the form of a uniform text and did not arouse controversy.
My reservations are for table 4, which is a nuisance to the reader.
I understand that the authors insisted on tables 3 and part of the bibliography is presented here. Table 3, the information contained in it, could be presented as another item in the article.
Please think again about these two tables.
Author Response
R:After re-examining the text, I would like to say that it has its advantages. A very good review article would emerge from this manuscript.
A:Thank you for this comment
R:I do not understand why the authors insist on presenting the views of other researchers in tabular form.
This presentation would be very nice in the form of a uniform text and did not arouse controversy.
My reservations are for table 4, which is a nuisance to the reader.
A:We have put the main results from the literature in table to make it easy for the reader to get an overview of important findings in the area. In the discussion we put the results into three different “approaches” and to give perspective of the findings results are then further discussed from different angels.
R:I understand that the authors insisted on tables 3 and part of the bibliography is presented here. Table 3, the information contained in it, could be presented as another item in the article.
Please think again about these two tables
A: Tables 3 and 4 may be put together, however that would be a very heavy table. That is why we choose to split them into two tables. We do not understand what is meant to put the information from table 3 as an “item”.

Reviewer 3 Report
The authors solved all the problems highlighted by the reviewers. I would have preferred the presence of the Conclusions section with strengths and limitations inside it and not a separate section, but it's okay anyway. Well done!
Author Response
R:The authors solved all the problems highlighted by the reviewers. I would have preferred the presence of the Conclusions section with strengths and limitations inside it and not a separate section, but it's okay anyway. Well done!
A: We have added a section “Concluding Discussion” which include conclusion, strengths and limitations.
